# Parametric Approach to Simplified Life Cycle Assessment of Social Housing Projects

Eleni Eleftheriou [1], Luis Felipe Lopez Muñoz [2], Guillaume Habert [1] and Edwin Zea Escamilla [1,*]

1   Chair for Sustainable Construction, Swiss Federal Institute of Technology ETHZ, 8092 Zürich, Switzerland; eleni.eleftheriou@alumni.ethz.ch (E.E.); habert@ibi.baug.ethz.ch (G.H.)
2   BASE-Bahay Innovation Centre, Makati 1231, Philippines; luis.lopez@base-bahay.com
*   Correspondence: zedwin@ethz.ch

**Abstract:** The provision of sustainable housing solutions is one of the main challenges in emerging economy countries. Furthermore, it is clear that a sustainable solution should be based on renewable bio-based materials. Scientific and practical evidence clearly suggests that the use of bamboo in the provision of housing solutions provides communities with both environmental and socio-economic benefits via this strategy. One barrier to the promotion of this type of solution is the lack of knowledge on structural design and environmental performance. Moreover, access to assessment tools and methodologies is limited. The use of simplified Life Cycle Assessment (LCA) has exhibited great potential to increase accessibility, but the generation of life cycle inventory data remains a major issue. In this paper, we describe the development of a methodological approach to use parametric design to generate the data required to carry out simplified LCA of social housing solutions. Moreover, we present a case study assessing a housing unit using cement bamboo frame technology developed by the Base Bahay Foundation in the Philippines. The main parameters for the LCA of the buildings were identified through sensitivity analysis. Moreover, they show that parametric design is a valid approach to overcome the challenges of data generation at early stages of design. The proposed approach would enable users without civil and/or engineering background to carry out simplified LCA calculations. Thus, through methodological approaches, it is possible to reduce significantly the complexity associated with LCA and open new avenues for it application.

**Keywords:** bamboo; $CO_2$; design; environmental impact; LCA



## 1. Introduction

One of the most important targets within Sustainable Development Goal 11 is the provision of dignified housing for all [1]. This target poses a very high challenge in emerging economy countries where a balance between affordability and delivery speed is of great relevance [2]. Moreover, the built environment is responsible for more than 40% of the global consumption of energy and one-third of the global greenhouse gas emissions and, therefore, exhibits the highest potential to achieve a significant reduction in the environmental footprint [3]. Population growth in urban areas together with natural disasters [4] further increases the demand for housing solutions [5]. Furthermore, the production of conventional construction materials, such as ceramic, steel, and cement, is related to high levels of the primary energy demand and $CO_2$ emissions [6]. Thus, it is crucial that the housing demand challenge is solved via the adoption of sustainable solutions, aiming to decrease $CO_2$ emissions originating from the production of construction materials [7]. A new regenerative approach should be established, according to which human activities do not deplete environmental sources [8]. This suggests that the housing provision challenge should ensure affordability and guarantee access to the resources needed [9,10]. This idea has led to the development of alternative low-carbon approaches based on bio-based materials, such as bamboo [9].

### 1.1. Bamboo Construction

Bamboo is a strong, fast-growing, and lightweight construction material [11,12]. Bamboo grows naturally in Africa, Asia, America, and Oceania [12], and due to the great versatility, culms of bamboo have been widely used in construction [13]. Bamboo can be used in low industrialized form such as bamboo poles, flattened bamboo, and woven bamboo mats, or in industrialized form such as glue-laminated bamboo and laminated woven bamboo mat panels [14]. Bamboo structures are used worldwide, especially in Africa, Asia, and Latin America. Concerning their structural behaviour, bamboo materials can cover large spans and achieve an excellent performance under compressive forces [15]. Nevertheless, due to their inherent geometric variability, the application in structures is limited in comparison to other industrial applications of bamboo [16].

In general, two types of bamboo-based structures can be identified based on their construction systems, spatial trusses, and frame walls. The first type can be used in open structures such as pavilions or bridges, as shown in the examples in Figure 1. These types of structures are the most common and easily recognized. The construction principles are the same between both construction systems, with the main difference representing the use of claddings in the frame wall system.

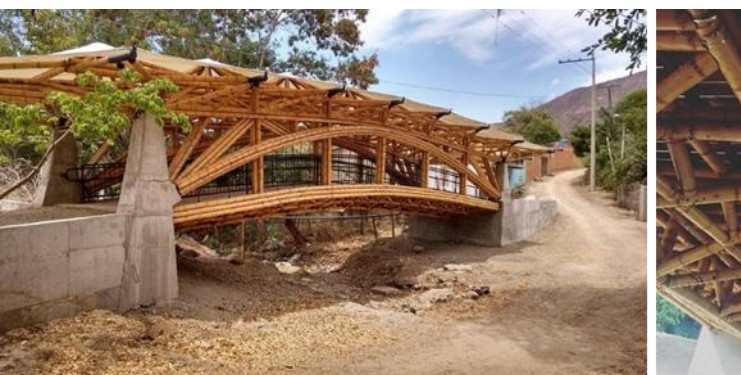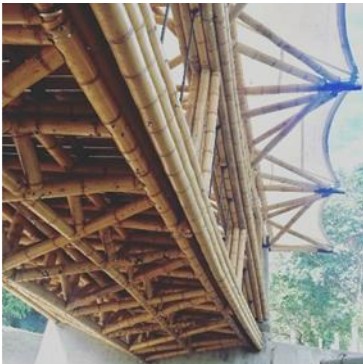

**Figure 1.** Low-traffic bamboo bridge (source: Authors).

As the name indicates, bamboo frames are used to create building shear walls. This construction system is commonly referred to as engineered bahareque, but regional differences can be found [17]. In 2002, the CEEA included two chapters regarding the structural design of bamboo structures in the Colombian building code [18], one chapter for spatial trusses and the other chapter for frame wall buildings up to two stories. This technology has been implemented globally and is now part of a series of ISO norms [19,20]. One example of this evolution is cement bamboo frame (CBF) technology. CBF is a localized version of the engineered bahareque system, which has been used in the social housing sector in the Philippines since 2013. The load bearing structure is constructed of round bamboo, the connections consist of steel and mortar, and the cladding entails a conventional plaster finishing layer, as shown in Figure 2.

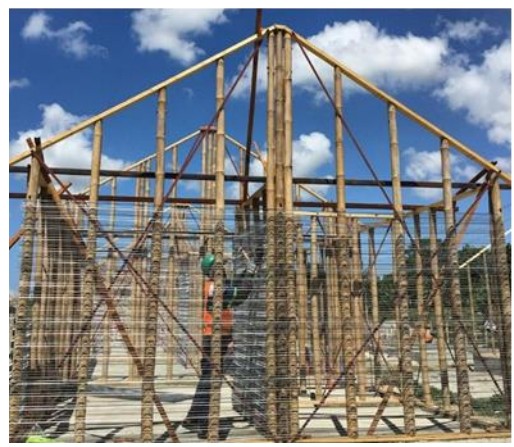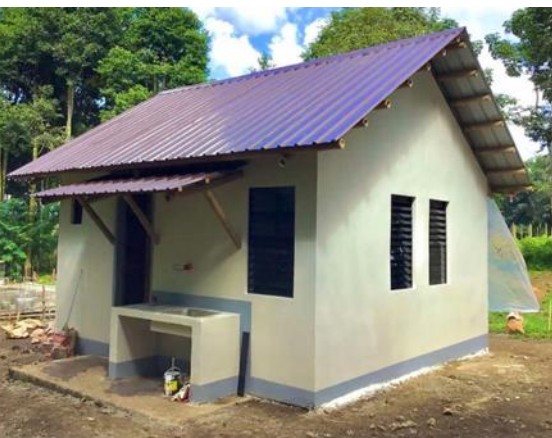

**Figure 2.** Single-family house using CBF technology in the Philippines (source: Base Bahay).

*1.2. Life Cycle Assessment (LCA)*

Life cycle assessment (LCA) is the main methodology used to quantitatively assess the environmental impacts related to the production and use of goods throughout their service life [21]. The LCA method is methodologically described in the ISO norm series 14040 [22] and in national standards worldwide. The life cycle of a building can be divided into four methodological phases: (i) the production and transportation of materials, (ii) the construction phase, (iii) the use/operational phase, and (iv) the demolition and disposal phase [23]. A general view of the life cycle stage information of buildings, based on the EN15978 norm, is shown in Figure 3. LCA can be summarized in four steps: goal and scope definition, life cycle inventory (LCI), life cycle impact assessment (LCIA), and interpretation of the results [24]. LCA results can be used to identify environmental hotspots of the studied system. Thus, this technique provides information for the decision-making process regarding the environmental impact products and processes [25]. Moreover, LCA can be used to compare alternatives of a production process or products. The application of LCA is hindered by high levels of complexity in terms of data generation [26] and allocation of impacts and characterisation of results [27]. Moreover, the lack of representative data is perceived as one of the greatest barriers to overcome [28].

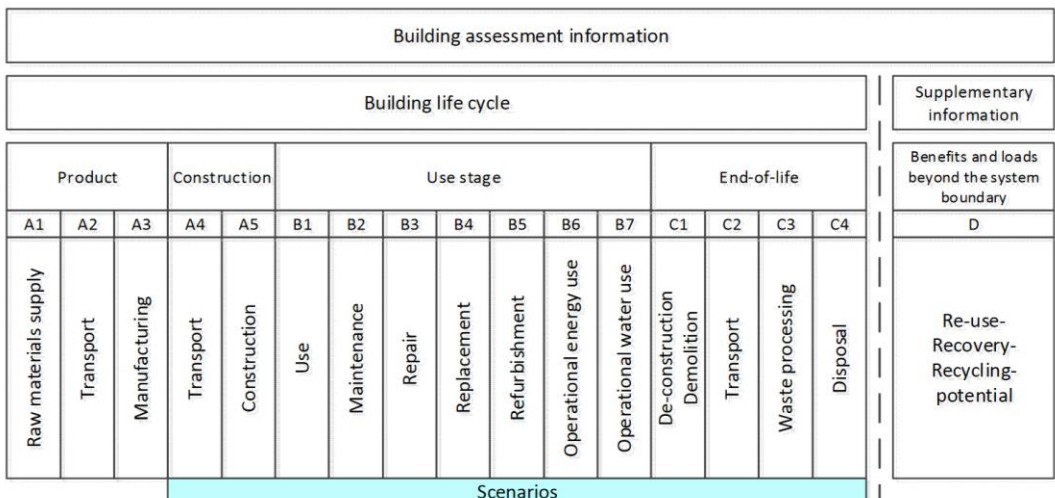

**Figure 3.** Building life cycle stages.

Currently, there exists a wide range of specialized software to perform LCA. These software packages range from licenced solutions to open-access programs. Moreover, there are both proprietary and open-source software packages. In general, all these software

solutions are stand-alone applications and are challenging to interface with computer-aided design software [29]. In addition, LCA software requires life cycle inventories (LCIs), which are usually found in databases. These are curated by many different actors, and access to these databases also varies, from licence-based to open-access formats. These databases can be topic-specific or multi-topic databases, and their coverage can range from country-specific to global values.

### 1.3. Parametric LCA

LCA is a data-driven methodology and requires detailed information, which is often not available at the early design stages. These stages impose the greatest influence on the environmental performance of buildings. LCA is often used as a post-ex assessment tool, which limits its application and significantly reduces the usefulness of its results [30]. Simultaneously, the application of LCA in the building process is hindered due to its inherent complexity and time demand. This limitation has driven the development of simplified approaches that could be used by different actors without the need for expert knowledge on LCA [28,31]. Among these approaches, parametric LCA has been identified as a promising solution. This approach is particularly interesting for Non-Governmental Organisations (NGOs) and humanitarian organizations operating in the housing sector, as this method allows users without LCA expertise to obtain results regarding the environmental performance of their designs.

Hollberg et al. described the main challenges and opportunities for parametric LCA [3,30,32,33]. In principle, the method should be easy to understand and widely applicable, especially at the early design stages. During the course of the project, any assumptions should be replaced with specific data [3]. Furthermore, the level of detail (LoD) should be determined, such as the building, building elements, element components, and component materials, which are used for the bill of quantities (BOQ) [33]. The integration of BIM and LCA is a developing field, which inherits the challenges of LCA related to poor accessibility of data and lack of compatibility of the data structures [34].

The current approaches still relay on the availability of data at early stages of design, in practice this data is not available. Moreover, when developing a housing project, the decisions regarding construction systems and materials are done before the actual designs are developed. These decisions make a significant difference on the carbon footprint of the final buildings. Thus, the need for solutions that allow non-experts to carry out LCAs at early stages of design is needed.

The objective of the work presented in this paper was to identify the set of parameters required to conduct simplified LCA of social housing units using both conventional and bamboo-based construction systems. Moreover, to develop a simplified approach to link the structural check of early-stage designs with the bill of material required to construct the proposed housing units.

### 2. Materials and Methods

The methodological approach developed was intended to obtain single-story social housing solutions. Figure 4 shows the methodology consisting of three interacting modules: (i) structural design module with green background; (ii) LCA module orange background, and (iii) user interface blue background. The latter module provides two main functions: the first function collects user inputs and communicates the results. The data can be collected at two detail levels, depending on the expertise of the user. At the basic level, the user inputs information regarding the dimensions of the studied building (length, width, and height) and the number of doors and windows. At the expert level, the collected information refers to structural design parameters such as soil types, concrete and steel grades and external forces such as earthquakes and winds. Regardless of the level of detail, a calculation can be performed.

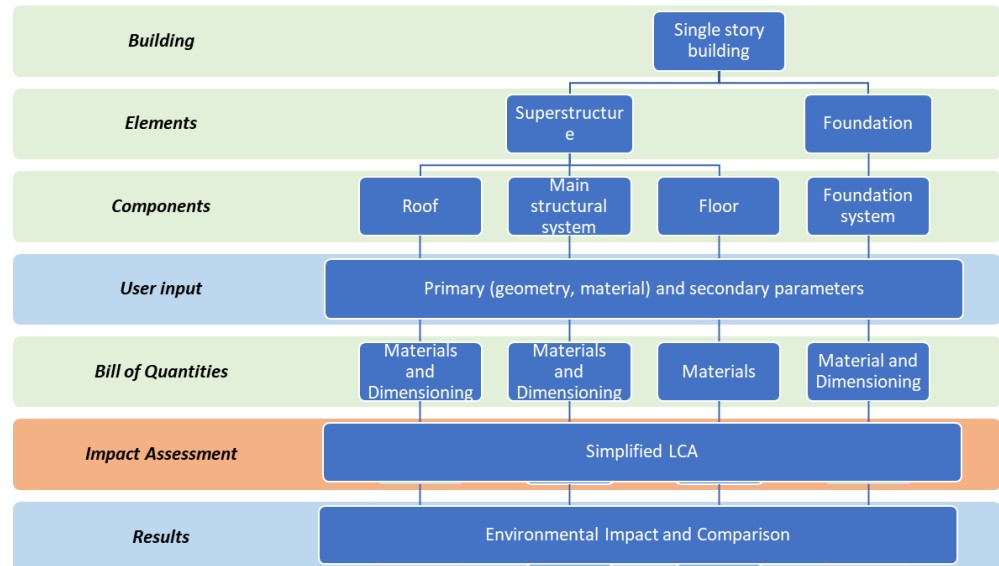

**Figure 4.** Conceptual framework.

The structural design module allows for the calculation of two construction systems: a concrete post and beam and bamboo frame walls. Once the construction type is selected by the user, the module performs structural calculations and designs building elements starting with the super-structure. These values are used to calculate the material amounts required for each specific construction system. In a parallel workflow, the module calculates the dead loads required for building foundation calculations. Depending on the level of details used, the module can adopt either default values (basic) or user data (expert) as inputs related to the soil type and seismic and wind loads.

## 2.1. Construction Systems and Materials

The main materials used in the design include reinforced concrete, concrete hollow blocks, cement mortar, steel, fired clay bricks, wood, and bamboo. The concrete strength is either 20 or 25 MPa. The steel quality includes B500, Grade 33, Grade 40, and Grade 60. The brick dimensions in mm are 190 × 90 × 60. The design values for the bamboo poles and timber are listed in Table 1.

**Table 1.** Material design values.

| Property | Symbol | Bamboo | Wood (50% Stress Grade) |
|---|---|---|---|
| Compression strength parallel to the grain | $f_{c,o,adm}$ | 8 | 6.54 |
| Bending strength | $f_{m,adm}$ | 7.7 | 10.3 |
| Shear strength | $f_{v,adm}$ | 8 | 1.69 |
| Tension strength parallel to the grain | $f_{t,o,adm}$ | 27 | - |
| Tension strength perpendicular to the grain | $f_{t,90,adm}$ | 0.09 | - |
| Modulus of elasticity—mean | $E_{mean}$ | 15,500 | - |
| Modulus of elasticity—5th percentile | $E_{min}$ | 13,500 | 2910 |

## 2.2. Input Parameters

The input parameters can be divided into principal (basic mode) and secondary parameters (expert mode). The principal parameters include the building perimeter, number of external doors, number of windows, height, number of interior rooms, and roof material. These parameters are input by the user in both modes. The secondary parameters include the soil quality, concrete quality, steel quality, wind loads, and earthquake loads. These parameters are only input if the user possesses the expertise to select the appropriate

values. In the basic mode, all the secondary parameters can be assigned default values, as summarized in Table 2.

**Table 2.** Default values for the secondary parameters.

| Material | | |
|---|---|---|
| **Concrete strength** | fc′ = 20 | MPa |
| **Steel quality** | Grade 40 | - |
| **Roof** | | |
| **Roof type—choice 1** | Gable | |
| **Roof angle** | 30 | degrees |
| **Foundation** | | |
| **f (length below the soil)** | 0.6 | m |
| **Soil permissible stresses** | 50 | |
| **Loads** | | |
| **Additional surcharge (concrete)** | 1 | kPa |
| **Additional surcharge (bamboo)** | 0 | kPa |
| **Roof live load** | 0.6 | kPa |
| **Input for Earthquake Force Calculations** | | |
| **Soil profile** | SC—very dense soil and soft rock | - |
| **Seismic zone** | 4—rest | - |
| **Seismic source type** | A—7 ≤ M ≤ 8.4 | - |
| **Input for Wind Force Calculations** | | |
| **Wind speed** | 250 | kph |
| **Exposure category** | B | - |

The user can determine the number of doors and windows, which exhibit standard dimensions. The number of exterior doors can either be one or two, and the maximum number of windows can reach two if there occurs one exterior door or four in the case of two doors. The number of interior doors matches the number of interior separations (rooms) defined by the user. If the number is zero, then only exterior walls exist. To consider openings, the following standard dimensions were considered: the typical door opening was 2.1 m high and 1 m wide, and the typical window opening was 1.2 m high and 0.9 m wide. It should also be pointed out that the height provided as input was the height of the CBF walls, not the height of the structure.

*2.3. Load Calculation*

The load calculation was based on the National Structure Code of the Philippines (NSCP 2015) [35]. Where allowable stresses or allowable strength design was used, i.e., for the design of the bamboo and timber members and soil stresses, the basic load combinations in Section 203.4.2 were used [35]. Where strength design or the load and resistance factor was used, i.e., for concrete members dimensioned under bending and shear, the load combinations in Section 203.3.1 were used [35]. The height was calculated at the subsequent stages by considering other parameters, such as the roof type and existence of concrete hollow blocks (CHBs) below the walls.

The earthquake forces were calculated according to Chapter 208. In bahareque houses, the main load-resisting system comprises bamboo walls, whereas in concrete houses, ordinary reinforced moment-resisting frames occur. The main difference between these two load resisting systems is that the structural period differs because the walled structure is stiffer. This results in a lower design base shear coefficient ($V_{max}/W$) value for concrete frames. The earthquake force ($V_{max}$) depends on the weight of the structure (W), so similar to the first step, only the $V_{max}/W$ coefficient is determined.

Wind load analysis was performed following NSCP-2015 [35] for enclosed or partially enclosed buildings, and the analytical procedure for low-rise buildings (Section 207C.4.1) was implemented. The possible load cases are shown in Table 3. The torsional load case was omitted in the calculations, given that the investigated buildings are low-rise buildings and the height is less than 9 m [35]. The wind loads depend on the geometry of the building. Therefore, a separate wind analysis was implemented of these two types of buildings. The self-weights of the structural and non-structural materials employed can be found in Table 3.

**Table 3.** Unit weights of the materials used.

| Structural Materials | Unit Weight | Unit |
|---|---|---|
| Concrete | 25 | kN/m$^3$ |
| Cement Mortar | 1900 | kg/m$^3$ |
| Steel | 7850 | kg/m$^3$ |
| Fired Clay Bricks | 741 | kg/m$^3$ |
| Wood | 5.4 | kN/m$^3$ |
| Bamboo | 7 | kN/m$^3$ |
| **Non-Structural Materials** | **Unit Weight** | **Unit** |
| G.I. Sheets | 0.0314 | kN/m$^2$ |
| 100-mm CHBs | 3 | kN/m |
| Riblath (per 20 sheets) | 25 | kg |

*2.4. Cement Bamboo Frame Building Structural Calculations*

The bamboo frame walls were designed to resist horizontal forces, such as winds and earthquakes. The shear capacity of a bamboo frame with one side of mortar plaster is 4 kN/m, whereas that of a bamboo frame with two sides reaches 8 kN/m. The total shear capacity of the structure along each direction is the shear capacity of the bamboo frame with two sides of mortar plaster times the sum of the effective length of the wall. The effective length of a two-sided wall equals the length of the wall, while that of a one-sided wall equals half the length of the wall. Based on this concept, the required amount of the effective wall length can be calculated so that the walls can resist the induced shear force, which is the maximum base shear due to winds or earthquakes. It was assumed that all the external walls are single-cladding walls unless the required wall length is greater than that provided by this assumption or the user desired to input more information. The length of the intermediate walls was assumed to equal the length of the building along each direction unless more information was provided. Based on the perimeter of the building, the height of the walls and the number of openings and presence of walls, the weight of the structure was assumed. The weight of the structure is required to determine the seismic and wind forces, which determine the design of the structure in regions where horizontal loads prevail. According to these forces, the minimum required length of the walls could be calculated.

*2.5. Concrete-Brick Building Structural Calculations*

The loads carried by the concrete frame depend on the material of the roof. The beams and columns were first dimensioned to carry the vertical loads according to the provisions of EN-1992-1-1:2004 [36]. At this stage, the minimum reinforcement was calculated so that the concrete does not fracture when the cracking moment is reached. In the next step, the beams and columns were again dimensioned in compliance with the capacity design requirements of EN-1998-1:2004 [36], since the horizontal forces are significant in the Philippines. Brittle failure should be prevented by deriving the design action effects in selected regions from equilibrium conditions, assuming that plastic hinges with possible over-strengths were formed in adjacent areas. The structure could be classified in the middle ductility class. Therefore, coefficient $\gamma_{Rd}$ equaled 1 for the beams and 1.1 for the columns. The minimum value of the mechanical volumetric ratio of the confining hoops

$\omega_{wd} = 0.08$ was provided within the critical region at the base of the primary seismic columns.

In the calculation of the vertical loads, the height of the beams was derived from the obtained data, whereas the dimensions of the columns were chosen so that the normalized design axial force $\nu_{Ed}$ does not exceed 0.3. This limitation was considered to perform an initial assessment of the cross-section. However, horizontal forces induce further limitations. The height of the cross-sections should resist the shear forces without exceeding the allowable stresses acting on the concrete diagonal. Furthermore, earthquake forces result in a reduction in the compressive force acting on the columns. In addition, given that a one-story building was considered, the compressive force acting on the columns was relatively low. Therefore, it could be assumed that the axial force given the earthquake combination was negligible, and structural verification could be performed accordingly. Additionally, it was required to maintain the depth of the compression zone below 45% of the static depth to achieve ductile behaviour. This resulted in a normalized acting moment of $\mu_{sd} = 0.295$. Finally, the dimensions of the columns were derived so that all the above-mentioned limitations were satisfied. The beam was finally reinforced with continuous bars, all along the length, according to the critical combination. The ductility requirements resulted in further constraints, such as the spacing of the stirrups, spacing of the longitudinal reinforcement, confinement of the critical regions and base of the columns, which were executed based on the provisions contained in EN-1998-1:2004 [36].

Once the structural calculations were carried out, the module could calculate the bill of materials for the foundation, superstructure, and roof. This information was then transferred to the LCA module to create the required life cycle inventories.

### 2.6. Simplified Life Cycle Assessment

In the LCA module, the functional unit was defined as a single-story building with an area input by the user not exceeding 64 $m^2$ and a wall height not exceeding 3 m. A minimum of one door and one window was also specified. The scope of LCA typically ranges from the determination of construction materials to the erection of buildings. These calculations do not include the biogenic $CO_2$ contributions of bio-based materials. The conducted LCA was geographically located in the Philippines, but this technique could be adapted to other geographic regions in the future.

Life cycle inventories were automatically generated using the user inputs and the bill of quantities generated in the structural design module. These life cycle inventories represent the production efficiencies and energy mix in the Philippines. The process of data characterization followed the methodologies developed by Zea and Habert [28] based on the Ecoinvent 3.6 database [37] and evaluation method of the IPCC (2013) [38]. With the use of this information and bill of quantities, a life cycle impact assessment was conducted. The calculation results were expressed in t $CO_2$-eq, which represents the main unit in the global warming potential impact category. The obtained LCIA results were presented at different disaggregation levels; this type of presentation allows a better understanding of the contributions of the different construction materials and/or building components. Moreover, the results could be stored and used to compare different construction systems, foundations, and/or roofs.

### 2.7. Case Study

To test the proposed approach, we used a case from the Base Bahay foundation being developed in Batangas (PH). This project consists of attached single family houses of ne storey of 27.3 $m^2$. A rendered view of the architectural design is presented in Figure 5. The primary and secondary parameters used for the calculations are presented in Table 4. In order to carry out a comparative assessment, the calculations were performed for CBF and concrete–brick construction systems. The same primary and secondary factors were maintained for both options.

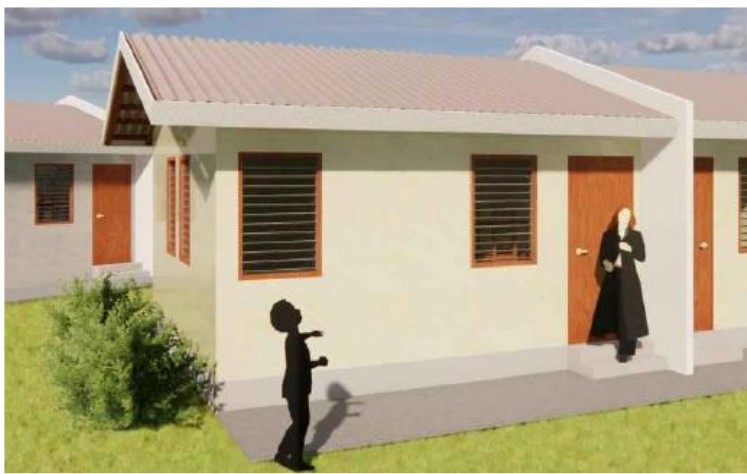

**Figure 5.** Housing unit type. Source: Base Bahay Foundation.

**Table 4.** Primary and secondary parameters case study.

| Primary Parameters | | |
|---|---|---|
| lx (parallel to building ridge—-if existent) | 5.7 | m |
| ly (normal to building ridge—-if existent) | 4.8 | m |
| Number of external doors (1 × 2.1) | 2 | - |
| Number of windows (0.9 × 1.2) | 5 | - |
| Minimum height (height of the walls) | 2.8 | m |
| Construction system | CBF/Concrete-Brick | |
| Interior separations | 5 | - |
| **Secondary Parameters** | | |
| Material | | |
| What is the concrete strength? | $fc' = 25$ | MPa |
| What is the steel quality? | Grade 30 | - |
| Roof | | |
| Roof Type—-Choice 1 | No slope | |
| Roof angle | 30 | ° |
| Foundation | | |
| f (length under soil) | 0.9 | m |
| Soil permissible stresses | 30 | |
| Loads | | |
| Additional surcharge (concrete) | 2 | kPa |
| Additional surcharge (bamboo) | 0 | kPa |
| Roof live load | 0.6 | kPa |
| Input for earthquake force calculation | | |
| Soil profile | SE-Soft Soil | - |
| Seismic zone | 4—rest | - |
| Seismic zource type | A—$7 \leq M \leq 8.4$ | - |
| Distance to seismic source (km) | 35.1 | km |
| Input for the wind's force calculation | | |
| Wind speed (Wind. Map Figure 207A.5-1A, B or C) | 340 | kph |
| Exposure category | B | |

## 3. Results and Discussion

The proposed approach allowed us to develop a robust yet user-friendly modular methodology to perform simplified LCA studies using parametric design while ensuring the structural performance of the studied buildings. The methodology was implemented, and Excel calculations were executed to assess the approach. A case study in the Philippines was used to evaluate the different features and sensitivity of the input detail level. Figure 6 shows the calculation results using basic and expert-level details. This figure shows that a

variation of $+/-8\%$ in the results can be expected. Nevertheless, this level of variability in the results is acceptable, and a compromise between the necessary data and expertise demands that a more precise result could be required. This figure also shows that the inputs in the expert mode can significantly influence the results. Thus, it could be advisable that the team working with this tool should involve at least one expert on structural design. As shown in this figure, the use of cement bamboo frame construction systems could achieve environmental savings of approximately 60% over conventional construction systems.

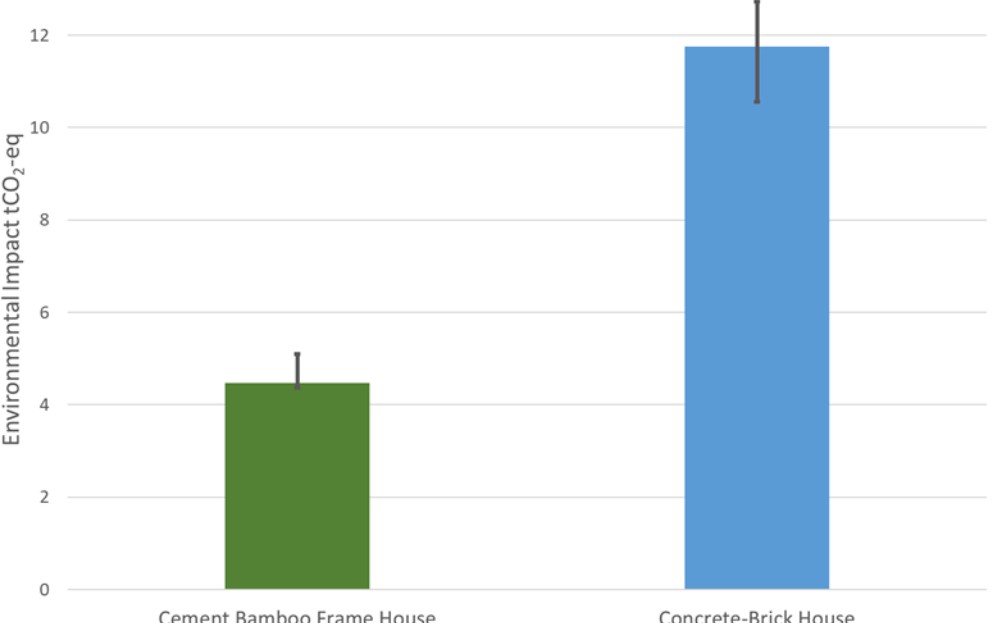

**Figure 6.** Comparative LCA.

The proposed approach also includes contribution analysis. Figure 7 shows the results for the bamboo-based house. This figure reveals that the foundation is the main component contributing to the building environmental impact, closely followed by the walls. From the outer layer in the figure, we can observe how concrete and steel and the main contributing materials affect each building component. These results highlight the importance of appropriate structural design not only to guarantee the safety of future inhabitants but also to maximize the environmental savings potential of the use of bio-based materials. Furthermore, these results emphasize the importance of the appropriate selection of materials and systems when making the case for sustainable construction.

To better identify the importance of the various parameters under each of the modes, sensitivity analysis was carried out. The sensitivity analysis results are presented in the following section, first evaluating the parameters under the basic mode, then those under the expert mode and finally providing an overall view of the parameters and their contributions. In these analyses, the maximum variations in each parameter and the results were recorded. With this information, the percentage of change in each parameter was defined.

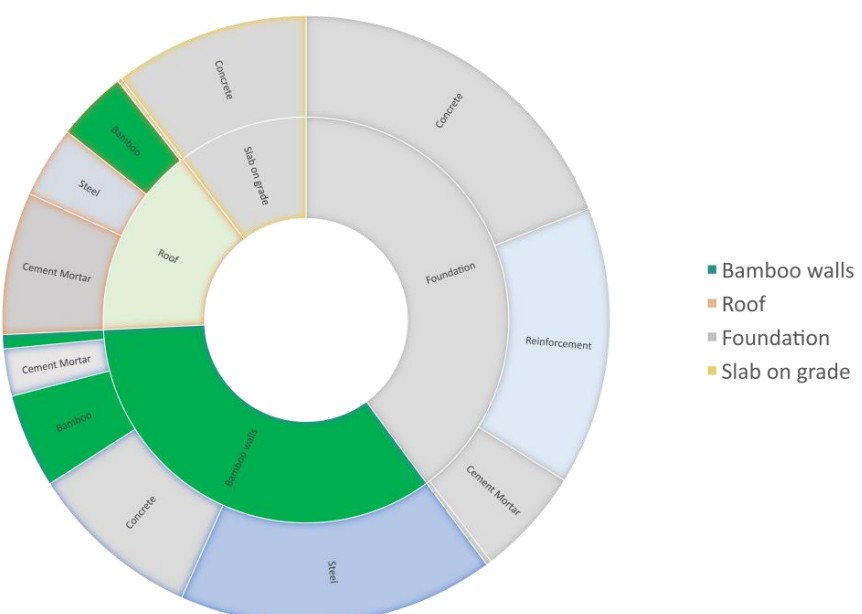

**Figure 7.** Contribution to the environmental impact at the building component and material levels.

### 3.1. Basic Parameters

The first parameter studied was the building perimeter. In this case, the length of the walls along the X (lx) and Y (ly) axes was the only parameter varied. This was achieved to avoid interference resulting from other parameters in the calculations. The results indicated that for any given value of lx, the environmental impact of both bamboo and concrete houses seemed to increase with increasing lx value, and consequently, the gross floor area increased. Additionally, the emissions increased with increasing ly value with respect to lx, as shown in Figure 8. The reason is that the weight and the vertical and horizontal actions also increased. Regarding the bamboo building, the observed increase mostly depended on the structural capacity of the walls along each direction, which linearly depended on the structural length of the walls. In terms of the concrete building, however, the shape of the increase curve tended to be parabolic. The reason for this is that the maximum acting moment increase was analogous to the square of the free-span length. The above moment increase resulted in higher reinforcement requirements, which increased the environmental impact.

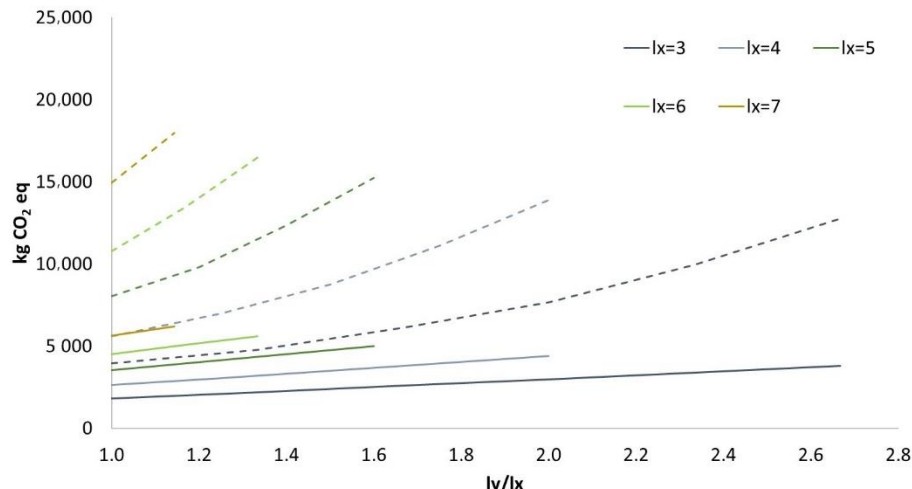

**Figure 8.** Variation in the total emissions with changes in the perimeter. Bamboo-based (solid) and concrete (dashed) structures.

The results indicated that the emission difference per gross floor area did not remain constant and, therefore, highly depended on the geometric parameters. This observation highlights the importance of parametric approaches to this kind of calculation.

Another basic parameter considered in sensitivity analysis was the number of doors and windows. In regard to this parameter, the total $CO_2$ emissions decreased with increasing number of doors and windows since less material was used. The change in emissions of the bamboo structure was nonsignificant compared to that in the emissions of the concrete structure. This could be explained based on the structure of a typical door and window bamboo panel, in which the outline of the opening is constructed of wood, which is connected with additional bolts and nuts. Regarding the concrete structure, the change was attributed to the large difference in the number of bricks. Consequently, the difference in the emissions of the bamboo-based concrete structure decreased with increasing number of openings, since the increase in the number of openings significantly reduced only the environmental impact of the concrete house. In the investigated example, the emission difference decreased by 10% with additional openings.

### 3.2. Advanced Parameters

As mentioned above, the user can choose to increase the level of detail of the input parameters, which requires more advanced knowledge and expertise. In this section, we present the results for five of the most relevant parameters in the expert mode. The first expert parameter studied was the quality of concrete. In this case, we observed a direct relation between the increase in concrete strength and the total environmental impacts. However, the difference in emissions between these two solutions changed by only approximately 1%. This difference increased when considering a flat concrete roof for the concrete building, reaching 8%.

The second parameter analysed was the quality of steel used for reinforcement. The analysis demonstrated that a lower steel quality could result in much higher emissions of the concrete structure and, therefore, in a much sharper increase in the total emissions. The reason is that the capacity of the concrete cross-sections highly depended on the maximum steel force. In the case of Grade 33 steel, the yield strength was 227.5 MPa, which is approximately half of the value of B500 used in Europe. Consequently, approximately double the reinforcement was required. The steel quality influenced only the foundation of the bamboo structure. Therefore, the difference was not pronounced. As a result, the difference between these two design scenarios could increase by 45%.

The third parameter analysed in the expert mode was the roof shape and material. The analysis results indicated that the roof shape significantly influenced the emissions. The reason is that the inclination of the roof raised the wall-material requirements. A mono-slope roof design could decrease the environmental impact of the roof, but this design could increase the environmental impact of the main structure. This configuration could require less roof material but more wall material. The concrete structure was more sensitive to the roof shape due to the high contribution to the total emissions of the walls and concrete columns. In regard to the observed difference, the value was greater for the mono-slope roof because, in this case, the increase in concrete was much more pronounced. These analyses indicated that if the roof material and angles remained consistent in both cases, a mono-slope roof could produce higher emissions than those under a gabled design.

The fourth parameter studied was the footing design. In general, the designed footing should be wide enough so that the permissible soil stresses are not exceeded. The investigated structures are one-story buildings, and, therefore, the weight was relatively small. Consequently, the permissible stresses could influence the emissions only if the weight of the materials increased or additional roof loads were added. The bamboo structure consists of lightweight materials. Therefore, the pressures are relatively low, and in most cases, the minimum foundation dimensions could be used. The results demonstrated that the environmental impact of the bamboo structure was not greatly influenced by the permissible soil stresses unless the permissible stresses remained below 30 MPa. The

opposite behaviour could be observed for the concrete structure, which relies on heavier materials. This resulted in higher axial forces acting on the columns, especially if the roof is constructed of concrete.

The final studied parameter entailed the external lateral loads in terms of winds and earthquakes. Several inputs are associated with this parameter, such as the soil types and seismic zones. In the first case, changes in soil profile selection did not produce significant changes in the results, as this factor impacted the seismic calculations. Thus, the material amounts for both structures did not require significant changes. In the case of seismic zones, we used the highest risk as a default value in all calculations. The change in seismic zone could generate a decrease in the total emissions as the material demand of both the foundation and superstructure of the buildings is directly related to these demands. In the case of lateral wind loads, the default value used in the basic calculations was 250 kph. The analysis results indicated that the influence of the wind load was less significant than that of the lateral loads stemming from earthquakes. In most cases, the critical design shear force is the seismic force, and the wind loads influence the roof. An increase in the wind speed increases the wind roof loads and, therefore, the environmental impact of the bamboo house.

The sensitivity analysis results are summarized in Table 5. In this table, the results for the inputs with the contributions to the variability in the results are higher than 6%. Based on this table, we can observe that the most important values are considered in the basic mode. This confirms the importance of appropriate design and material selection. Moreover, most inputs in the expert mode could yield very low contributions to the variability. It should be noted that these parameters were also considered in the basic mode calculations, but only default values were used. These parameters are fundamental to properly perform the calculations, and their absence could render the calculation results unreliable. The results indicated that a tool using only basic mode inputs and default values for expert mode parameters could produce reliable results without requiring a user with a high degree of expertise on structural engineering. Nevertheless, to achieve higher-quality results, it would be advisable to obtain the support of a structural engineer in the process and to define all parameters to the best ability of the users.

**Table 5.** Parameter sensitivity and variability.

| Parameter | Contribution to the Variability | Difference Variation |
|---|---|---|
| Geometry (lx, ly) | Very high | 90% |
| Concrete roof type | Very high | 95% for large geometries |
| Steel quality | High | 45% |
| Interior walls (rooms) | Moderate | 6–10% |
| Doors and windows | Moderate | 10% |
| Height | Moderate | 15% |
| Concrete quality | Moderate | 8% |

## 4. Conclusions

Based on the experiences accumulated during the development of the research work presented in this paper, we can conclude that the use of parametric design is a valid approach to the challenge of data generation for LCA purposes. The proposed approach allows bridging the knowledge gap both in structural design and LCA at early stages of design. This shows that this kind of information can be generated at relevant stages of the design process and support an informed decision-making process concerning materials and construction systems. The case study helps identifying that the most relevant parameters are related to the geometry of the buildings. The information required for this parameter is available at the earliest stages of conception of a social housing project. Moreover, the low data requirement to carry out these kind of calculation signals the potential application of this type of approach in the social housing sector. Nevertheless, further research is required, especially in the estimation transport distance of construction materials. Furthermore,

the integration of calculations regarding biogenic $CO_2$ stored in bio-based construction materials needs to be addressed in order to obtain a better picture of the environmental benefits from the use of bio-based construction materials like bamboo. Finally, we can conclude that the proposed approaches can be further developed into tools and applications that can benefit not only researchers but can also benefit many NGOs operating in the field of affordable sustainable housing worldwide. Thus, bridging the existing knowledge gap and promoting a science-based decision-making process in the sector.

**Author Contributions:** Conceptualization, E.E., G.H. and E.Z.E.; Data curation, E.E. and L.F.L.M.; Formal analysis, E.E.; Methodology, E.Z.E.; Supervision, G.H. and E.Z.E.; Validation, L.F.L.M.; Writing—original draft, E.E., L.F.L.M. and E.Z.E.; Writing—review & editing, L.F.L.M., G.H. and E.Z.E. All authors have read and agreed to the published version of the manuscript.

**Funding:** This research received no external funding.

**Acknowledgments:** The authors of this paper wish to acknowledge the invaluable support of the Base Bahay Foundation and HILIT Foundation regarding the development of the work presented.

**Conflicts of Interest:** The authors declare no conflict of interest.

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
