# Peer review of "Parametric Approach to Simplified Life Cycle Assessment of Social Housing Projects"

_sustainability, doi:10.3390/su14127409_

Round 1

Reviewer 1 Report

The paper presents a good topic related to. Parametric Approach to Simplified Life Cycle Assessment of  Social Housing Projects. The following comments and attached file to improve the paper.

Author Response

Our comments are on the file attached

Reviewer 2 Report

My comments for this paper are listed as follows:

  1. The abstract contains too much background introduction. The background introduction needs to be shortened and this paper’s contribution needs to be highlighted.
  2. The full name of NGO needs to be provided.
  3. Why isn’t LCA included in the keywords? Footprint should not be a keyword.
  4. The introduction section is too long. The background introduction needs to be shortened.
  5. The introduction section does not state the current gap clearly. It is also insufficient about this paper’s contribution, which is to solve the gap.
  6. Figure 4 needs to provide the meanings for different colors.
  7. Figure 4 needs to describe the three modules ((i) structural design module; (ii) LCA module and (iii) user interface). The current version does not show this.
  8. There is a typo for “Impact Assessment”.
  9. In Lines 245 to 247, does this mean that the principal parameters are not included in the secondary parameter list, but they are included in the expert mode? If yes, the first sentence of this paragraph needs to be modified; if no, this sentence needs to be modified to make it clearer.
  10. In Table 2, the default values for walls are not given. The values need to be calculated based on the actual situation. Thus, they should not appear here since this table aims to provide some default values.
  11. In Line 264, the NSCP should be the National Structure Code of the Philippines, not the National Building Code of the Philippines.
  12. Equations for the calculations need to be provided instead of just referring to the code sections.
  13. There are some errors for the references (Error! Reference source not found.).
  14. The description about the case is needed (e.g., the description of the studied building).
  15. This paper provides a case study for a certain building. I do not know how this result can help for other buildings.

Author Response

our comments are on the attached file

Reviewer 3 Report

The paper “Parametric Approach to Simplified Life Cycle Assessment of Social Housing Projects” describes the development of a methodological approach to using parametric design to generate the data required to carry out simplified LCA of bamboo-based buildings and presents a case study assessing a housing unit using cement-bamboo frame technology developed by the Base-Bahay NGO in the Philippines. After reviewing it, I have comments and suggestions as follows:

1). The current abstract is quite long; I suggest revising it summatively.

2). It seems insufficient when having only “CO2; footprint; bamboo” as keywords for your article. I suggest adding two or three more keywords.

3). In lines 38-39, you stated that “One of the most important sustainable development goals is the provision of dignified housing for all [1].” As we know there are 17 sustainable development goals (SDGs), and none of them is named the provision of dignified housing for all. I guess you referred to SDG 11 “Sustainable Cities and Communities (Make cities and human settlements inclusive, safe, resilient, and sustainable)” In this goal, affordable housing is one of the targets. So, what you stated is just one target of SDG 11. Hence, I suggest revising your statement accordingly.

4). Your introduction is quite long, which includes a literature review and several figures. I suggest revising this section by dividing it into “1. Introduction and 2. Literature Review”.

In section 1 (introduction), I suggest presenting state of the art (clarifying research problems, linked to your research questions or objectives). The significance of your research can also be presented in this section. Furthermore, avoid presenting the information that makes your state of art confused or unclear. For example, in lines 53-54, you mentioned alternative low-carbon approaches based on bio-based materials, bamboo and timber [9], but after that, you presented/chose only bamboo, without talking about the timber. So, in this example, just talking about bamboo is enough, don’t mention timber.

In section 2 (literature review), I suggest moving all literature review from the current introduction to this section, as well as providing a view on the disadvantage of bamboo materials in general, excluding life cycle, in order for you to discuss the findings well and make a good conclusion at the end. In this section, you can expand the literature of your research, because currently, you have only 30 references.

5). In the Introduction I see you discussed the reasons for using LCA methods; if possible, I suggest presenting this aspect in the methods section as method/tool selection.

6). As the results section is 1 page and the discussion section also includes some results, I suggest combining these two sections into one.

7). I suggest improving the content of and strengthening the conclusions.

Author Response

our comments are on the attached file

Round 2

Reviewer 2 Report

There are three previous comments that have not been replied. They are listed below:

1. The full name of NGO (non-governmental organization) [Line 127] needs to be provided.

2. Equations for the calculations need to be provided instead of just referring to the code sections [Line 267].

3. This paper provides a case study for a certain building. I do not know how this result can help for other buildings (Deeper discussion needs to be added in Section 3 or separate Section 3 into two sections – section for results and section for discussion).

In addition, there are two further comments:

1. In Line 14, both full name and the abbreviation of Life Cycle Assessment need to be provided.

2. The background introduction is too long and needs to be shortened.

Author Response

Dear Reviewer, you find our reply to your comments on the attached file

thank you

Reviewer 3 Report

The authors have not addressed Comment 4 yet. Based on this comment, I suggest expanding/strengthening your literature review.

Author Response

Dear Reviewer

Below our reply to your comment

Comment 1: The authors have not addressed Comment 4 yet. Based on this comment, I suggest expanding/strengthening your literature review.

 Reply: We did updated the text according to your recommendation and added several recent references. It is worth noting that the other reviewers are requesting to reduce this section, we are trying to find a balance point where we comply with these opposite requests.

Round 3

Reviewer 2 Report

The authors addressed all my concerns. I do not have any further comment.

Author Response

We thank you for your support 

Reviewer 3 Report

I read Reviewer 2's comment. I see that he or she also commented that "The background introduction is too long and needs to be shortened" while I had the same comment, but suggested separating the literature review and those figures in the 'Introduction' section to the adding (2. Literature Review) section (comment 4). However, I couldn't see any changes in this version compared to Round 2 while the literature review and those figures are still not separated from the 'Introduction' section to the "Literature Review" section, which means the current introduction section is still too long (my comment 4 is still not addressed).

Author Response

Dear Reviewer

We apologize for not replying directly to your comment. We consider that the state of the art was addressed on our document and we included several recent publications on the field on our review of the document. Moreover, we consider that for an article dealing with a methodological approach, like ours, an introduction discussing the state of the art is the most appropriate approach to present this information.

we thank you for your support during this process